# Neuroprotection of Radiosensitive Juvenile Mice by Ultra-High Dose Rate FLASH Irradiation

**DOI:** 10.3390/cancers12061671

**Published:** 2020-06-24

**Authors:** Yasaman Alaghband, Samantha N. Cheeks, Barrett D. Allen, Pierre Montay-Gruel, Ngoc-Lien Doan, Benoit Petit, Patrik Goncalves Jorge, Erich Giedzinski, Munjal M. Acharya, Marie-Catherine Vozenin, Charles L. Limoli

**Affiliations:** 1Department of Radiation Oncology, University of California, Irvine, CA 92697, USA; yalaghba@uci.edu (Y.A.); sammie1cheeks@gmail.com (S.N.C.); bdallen@uci.edu (B.D.A.); pmontayg@hs.uci.edu (P.-M.G.); ngoclied@uci.edu (N.-L.D.); egiedzin@uci.edu (E.G.); macharya@uci.edu (M.M.A.); 2Laboratory of Radiation Oncology, Department of Radiation Oncology, Lausanne University Hospital and University of Lausanne, 1000 Lausanne, Switzerland; Benoit.Petit@chuv.ch (B.P.); Patrik.Goncalves-Jorge@chuv.ch (P.G.J.); 3Institute of Radiation Physics/CHUV, Lausanne University Hospital, 1000 Lausanne, Switzerland

**Keywords:** FLASH radiotherapy, juvenile mice, cognitive dysfunction, neurogenesis, medulloblastoma, pediatric brain cancer, updating task, memory consolidation

## Abstract

Major advances in high precision treatment delivery and imaging have greatly improved the tolerance of radiotherapy (RT); however, the selective sparing of normal tissue and the reduction of neurocognitive side effects from radiation-induced toxicities remain significant problems for pediatric patients with brain tumors. While the overall survival of pediatric patients afflicted with medulloblastoma (MB), the most common type primary brain cancer in children, remains high (≥80%), lifelong neurotoxic side-effects are commonplace and adversely impact patients’ quality of life. To circumvent these clinical complications, we have investigated the capability of ultra-high dose rate FLASH-radiotherapy (FLASH-RT) to protect the radiosensitive juvenile mouse brain from normal tissue toxicities. Compared to conventional dose rate (CONV) irradiation, FLASH-RT was found to ameliorate radiation-induced cognitive dysfunction in multiple independent behavioral paradigms, preserve developing and mature neurons, minimize microgliosis and limit the reduction of the plasmatic level of growth hormone. The protective “FLASH effect” was pronounced, especially since a similar whole brain dose of 8 Gy delivered with CONV-RT caused marked reductions in multiple indices of behavioral performance (objects in updated location, novel object recognition, fear extinction, light-dark box, social interaction), reductions in the number of immature (doublecortin^+^) and mature (NeuN^+^) neurons and increased neuroinflammation, adverse effects that were not found with FLASH-RT. Our data point to a potentially innovative treatment modality that is able to spare, if not prevent, many of the side effects associated with long-term treatment that disrupt the long-term cognitive and emotional well-being of medulloblastoma survivors.

## 1. Introduction

Radiation therapy (RT) is a critical component in the treatment of medulloblastoma (MB), which is the most common type of primary brain cancer in children [1]. The overall survival rate of MB patients after receiving a combination of surgery, RT and chemotherapy routinely exceeds 80% [1,2,3]. Despite these favorable prognoses, pediatric survivors of MB show debilitating and irreversible radiation-induced neurocognitive decrements and mood disorders [4,5,6]. Moreover, they are prone to cerebral microbleeds, along with a two-fold increased risk of stroke, resulting from their intensive cranio-spinal irradiation (IR) regimens [7,8]. The prevalence and persistence of these toxic radiotherapy side effects point to the deficiency of current therapies, where the resultant neural toxicities, stemming from the exquisite sensitivity of the juvenile brain to radiation exposure, define a distinct clinical challenge. These prolonged and debilitating disruptions to neurocognitive health are now recognized as a major criterion for evaluating therapeutic outcomes and for determining long-term quality of life [9,10,11].

To address these unmet medical needs, our group has championed the usage of ultra-high dose rate FLASH-radiotherapy (FLASH-RT), an innovative ultra-high dose rate modality which has been shown to significantly limit normal tissue toxicity when compared to the conventional dose rate (CONV) irradiation regimens that are used in current clinical practice. FLASH-RT was originally defined by mean dose rates in excess of 40–100 Gy/s [12,13], an oversimplification that has now been recognized, and it has subsequently been re-defined by beam parameters that are able to provide intra-pulse dose rates on the order of 10^6^ Gy/s and able to deliver prescribed total doses in under one tenth of a second (i.e., total doses ≤ 0.1 s). Recent investigations, both by ourselves and by others, have pointed to a wide range of beneficial normal tissue sparing in multiple organs and preclinical models [12,13,14,15,16,17,18,19]. To date, these in vivo data sets remain the only bona *fide* measures available to substantiate and rigorously validate the occurrence of the “FLASH effect”.

As alluded to above, the typical RT protocols lead to deficits in learning and memory, attention, executive function, and a multitude of mood disorders that can be linked to hippocampal and cortical based alterations [20,21,22]. Many of these detrimental functional outcomes can be tied to a range of associated pathologies that are temporally coincident with reductions in dendritic arborization and spine density, vascular abnormalities, decreases in microvascular density, myelination, and significant increases in neuroinflammation [15,23,24,25,26]. It is our contention that some (if not all) of these radiation-induced injury signatures are contributory (if not causal) to the long-term manifestation of cognitive impairments. Recently, we have shown that FLASH-RT leads to a dramatic normal tissue sparing capability in the adult mice which have been subjected to whole brain irradiation when compared against isodoses delivered at conventional dose rates (~0.01 Gy/s) [15]. In adult mice, FLASH ameliorated short (1 month) and long-term (6 month) deficits in cognition, preserved host neuronal morphology, and attenuated astrogliosis and neuroinflammation [15].

The foregoing studies were undertaken in tumor free animals in order to avoid the confounding indications that are associated with the presence and growth of a tumor. It was uncertain from these findings, however, whether FLASH-RT could afford similar neuroprotective benefits within the more radiosensitive juvenile (3-week old) brain. To avoid any overt toxicity, and to use a dose known to elicit neurocognitive complications, we selected a single head-only dose of 8 Gy to conduct these proof of principle studies. While this dosing scheme is not common practice for the treatment of MB, it does constitute a typical hypofractionation dose that trends with many current RT protocols. Current findings have now provided a compelling evidence that young wild type animals subjected to high single dose FLASH-RT exhibit no radiation-induced cognitive impairments over protracted (2–4 month) post-irradiation intervals, along with minimal collateral radiation injury to the normal brain when compared to animals receiving equivalent conventional dose rate irradiation protocols. The present findings also provide the first piece of evidence that improved behavioral performance can be linked to the preservation of the neurogenic niche. These findings are remarkable, especially given the exquisite sensitivity of the juvenile brain to radiation-induced toxicities, and they point to the exciting prospect of implementing FLASH-RT in efforts to provide some long-awaited relief for children who are slated to undergo radiotherapy for the treatment of medulloblastoma.

## 2. Results

The goal of this study was to investigate whether FLASH-RT would result in similar cognitive sparing in the young mice as described in studies which were published previously in adult mice [15]. To this end, we utilized behavioral tasks with a specific emphasis on the hippocampal-medial prefrontal cortex neural circuits. Collectively, we show that FLASH-RT results in marked neuroprotective properties compared to conventional dose rate irradiation in young mice.

### 2.1. FLASH-RT Preserves Memory Updating in OUL over Time

The objects in updated location (OUL) task is a novel memory updating paradigm that is able to assess both the original memory and the updated information in a single test session [27]. Furthermore, OUL uses incidental learning that takes advantage of rodents’ innate preference for novelty. We first investigated the impact of FLASH-RT on the OUL task (Figure 1A) at two different timepoints: 2 and 4 months after irradiation. Following habituation, mice were first trained to learn the locations of two identical objects in a familiar context (training session, days 1–3). The next day, during the update session (day 4), all animals were exposed to one familiar object location (A_1_) and one object moved to a new location (A_3_). Control animals successfully acquired the original object location memory during the update session at both 2 and 4 months post-RT (Figure 1B), but mice irradiated with CONV-RT showed a significantly lower discrimination index (DI) compared to controls at both 2 (one-way ANOVA: F_(2,42)_ = 7.05, *p* = 0.0023; *p* = 0.0027, *n* = 15 per group) and 4 months post-IR (one-way ANOVA: F_(2,43)_ = 9.98, *p* = 0.0003; *p* = 0.0003, CONV: *n* = 15, Control: *n* = 16). Animals irradiated with FLASH-RT, on the other hand, failed to learn the update task at 2 months post-RT (*p* > 1.0, *n* = 15 per group), but were able to successfully learn it at 4 months post-RT (*p* = 0.0019, CONV: *n* = 16, FLASH: *n* = 15) (Figure 1B). These data demonstrate that mice exposed to FLASH-RT successfully perform memory updating in the OUL task over time, whereas CONV animals failed at both earlier and later timepoints.

The day after the update session, all groups were given a test session (day 5) to assess the animals’ recall memory for both the original object locations and the updated location. For the test, mice were exposed to four identical objects: three in previously exposed locations (A_1_, A_2_ and A_3_) and one in a novel location (A_4_). Memory for the updated information was examined via comparison between the exploration of the novel object location (A_4_) and the exploration of the updated location (A_3_). Memory for the original training was assessed by comparing the exploration of the novel object location (A_4_) to the exploration of the training object locations (A_1_). As mice prefer novelty, memory for either the original training session or the updated information is demonstrated by increased exploration of the object in the novel location (A_4_) compared to each of the other objects (indicated by a higher score on the DI (see methods).

Control animals show intact memory for the updated information 2 and 4 months post-RT (Figure 2A), but mice irradiated with CONV-RT showed a significantly lower DI compared to controls at both 2 (one-way ANOVA: F _(2,43)_ = 5.45, *p* = 0.0078; *p* = 0.004, *n* = 15 per group) and 4 months post-RT (one-way ANOVA: F_(2,44)_ = 11.47, *p* < 0.0001; *p* < 0.0001, CONV: *n* = 16, Control: *n* = 15). As for animals irradiated with FLASH-RT, this group failed to acquire the updated information at 2 months post-RT (no difference between CONV and FLASH-RT groups: *p* = 0.30, CONV: *n* = 15, FLASH: *n* = 16), but showed a significantly greater DI at 4 months post-RT compared with CONV (*p* = 0.0014; *n* = 16 per group) (Figure 2A).

With regards to the original information, these data were nearly identical to the updated information. Again, while control animals acquired the original information at both 2 and 4 months post-RT (Figure 2B), animals irradiated with CONV-RT showed impairments at both timepoints: (2 months post-RT: one-way ANOVA: F_(2,42)_ = 4.10, *p* = 0.024; *p* = 0.019; *n* = 15 per group; 4 months post-RT: one-way ANOVA: F_(2,42)_ = 13.04, *p* < 0.0001; *p* < 0.0001; *n* = 15 per group). Again, as with the updated information, animals irradiated with FLASH-RT failed to acquire the original information at 2 months post-RT (*p* = 0.078; *n* = 15 per group), but were able to successfully learn it at 4 months-post RT (*p* < 0.0001, *n* = 15 per group) (Figure 2B). Together, these results show that animals given FLASH-RT successfully learned and recalled both the updated and original information at the early (6 weeks) and protracted (12 week) post-irradiation times, but mice subjected to CONV-RT showed impairments in these memory processes at both 2 and 4 months post-RT.

### 2.2. FLASH-RT Minimizes Radiation-Induced Novel Object Memory Impairments as Well as Anxiety-Like Behaviors

Following the second OUL task, animals were tested sequentially on the following tasks: novel object recognition (NOR), light-dark box (LDB) and social interaction test (SIT). All animals were tested 4 months post-RT. Mice were first tested to determine whether FLASH-RT spares impairments in object recognition memory using the NOR task. Analysis of novelty using the DI for NOR shows that animals exposed to FLASH irradiation were statistically indistinguishable from the controls. However, the CONV-irradiated group exhibited a significant reduction in their DI compared to the controls (Figure 3A, one-way ANOVA: F_(2,40)_ = 7.072, *p* = 0.0023; *p* = 0.0049, CONV: *n* = 15, Control: *n* = 14) and FLASH-RT mice (*p* = 0.0048, CONV: *n* = 15, FLASH: *n* = 14).

Radiation exposure has been found to alter mood [28,29,30,31]. To determine the effects of FLASH and CONV irradiation on anxiety-like and social behaviors, mice were administered the LDB test and the SIT. The LDB test contrasts the natural inclination of mice to explore novel environments with their aversion to brightly lit spaces, thus evaluating their increased anxiety levels based on their greater preference for the dark versus light compartments of the testing arena [32]. Compared to CONV-irradiated animals, mice exposed to FLASH-RT showed a significantly greater number of light-dark transitions (one-way ANOVA: F_(2,44)_ = 3.67, *p* = 0.033; *p* = 0.0038, CONV: *n* = 16, FLASH: *n* = 15), a finding that did not extend, however, to time spent in the light compartment (one-way ANOVA: F_(2,44)_ = 1.50, *p* = 0.23, *n* = 15–16 per group) (Figure 3B).

Next, we used the SIT to examine social interaction behaviors. Social interaction behaviors are found to depend on hippocampal and medial prefrontal cortical circuits [33,34]. Mice that had been previously habituated with cage mates were allowed to interact with a novel mouse in a barrier-free arena. The total time that the test mouse spent interacting with the novel mouse or actively avoiding social interactions initiated by the novel mouse were recorded, following established protocols [35,36,37]. FLASH-irradiated animals spent a similar amount of time socially interacting with the novel mouse as well as avoiding the novel mouse. However, the CONV-irradiated mice spent significantly less time interacting with the novel mouse compared to the controls and the FLASH animals, respectively (one-way ANOVA: F_(2,43)_ = 4.47, *p* = 0.017; *p* = 0.037, CONV: *n* = 15, Control: *n* = 16; *p* = 0.019, *n* = 15 per group) (Figure 3C). Furthermore, CONV-irradiated animals spent significantly more time avoiding the novel mouse compared to the controls (one-way ANOVA: F_(2,40)_ = 6.83, *p* = 0.003; *p* = 0.003; CONV: *n* = 15, Control: *n* = 16) and the FLASH-irradiated animals *p* = 0.01; *n* = 14 per group) (Figure 3C).

Lastly, we examined whether FLASH-RT might preserve fear extinction (FE) memory, an active process that involves unlearning an association between a previously acquired conditioned response (tone) and an aversive unconditioned stimulus (foot shock), an indicator of memory consolidation. In the conditioning phase, all groups were given three tone-shock pairings (T_1_–T_3_; 120 s, 16 kHz tone followed by a 1 s, 0.6 mA shock; 2 min interval) and displayed a comparable percentage of time spent freezing during fear acquisition (Figure 4A, two-way ANOVA conditioning by group interaction: F_(4,86)_ = 1.16, *p* = 0.34; main group effect: F_(2,43)_ = 1.87, *p* = 0.17). During subsequent fear extinction trials in a new context (20 tones/day, 2 min interval, no shock), control and FLASH-irradiated mice exhibited a gradual decrease in freezing behavior over the course of the extinction training. In contrast, CONV-irradiated animals showed significantly higher freezing behaviors (days 1–3, data show response averaged over first set of 10 tones; (two-way ANOVA main group effect: F_(2,42)_ = 7.26, *p* = 0.002; Day 1: post hoc comparing CONV vs. FLASH: *p* = 0.032; Day 2: post hoc comparing CONV vs. FLASH: *p* = 0.0004; Day 3: post hoc comparing CONV vs. FLASH: *p* = 0.015; Figure 4A).

On the final day, an extinction test was given in which only three tones were administered (2 min interval). Control and FLASH-irradiated mice showed extinction of fear as indicated by a decrease in the percentage of freezing time (Figure 4B). In contrast, however, the CONV group showed persistent impairment (one-way ANOVA: F_(2,43)_ = 4.92, *p* = 0.012; *p* = 0.027, CONV: *n* = 15, Control: *n* = 15, FLASH: *n* = 16; Figure 4B). In sum, cognitive data which was collected 4 months following the exposure of juvenile mice provide compelling evidence for the long-term neurocognitive sparing of FLASH-RT compared to CONV irradiation.

### 2.3. FLASH-RT Spares the Neurogenic Niche in Juvenile Mice

Juvenile sensitivity to irradiation can, in part, be attributed to higher fractions of radiosensitive stem and progenitor cells within a given tissue [38]. Important in this regard is the capability of FLASH-RT to spare the neurogenic niche as quantified by the unbiased stereology of immature neurons (doublecortin positive cells, DCX^+^; Figure 5), as well as the percentage of cells which are dual-labeled for bromodeoxyuridine (BrdU) and a mature neuron-specific nuclear antigen (NeuN) in the hippocampal dentate gyrus (Figure 6). Compared to CONV-irradiated cohorts, the yield of DCX^+^ cells following FLASH-RT was statistically similar to the controls. Juvenile mice given CONV irradiation experienced a significant depletion of immature DCX^+^ neurons, an effect not found after FLASH-RT (one-way ANOVA: F_(2,9)_ = 8.05, *p* = 0.0099; *p* = 0.033: CONV vs. FLASH; *p* = 0.008: CONV vs. Control; *n* = 4 animals per group) (Figure 5).

### 2.4. FLASH-RT Preserves Neurogenesis in Juvenile Mice

To undertake a more formal assessment of neurogenesis, the number of NeuN positive mature neurons co-labeled with the thymidine analog BrdU was quantified along the subgranular zone of the denate gyrus. The one-way ANOVA revealed an overall significant effect for the percentage of BrdU^+^-NeuN^+^ cells F_(2,14)_ = 5.16, *p* = 0.021, and post hoc tests showed that both unirradiated controls (0 Gy) and FLASH-RT animals had significantly greater double-positive labeled cells than CONV-irradiated animals (*p* = 0.028: CONV vs. FLASH; *p* = 0.036: CONV vs. Control; *n* = 4–6 animals per group) (Figure 6).

### 2.5. Attenuation of Activated Microglia by FLASH-RT

To ascertain whether FLASH could attenuate microglia activation in juvenile mice subjected to CONV versus FLASH-RT, the number of IBA1^+^ (resting or total) and CD68^+^ (activated) microglial cells per animal was quantified in the hippocampus and expressed as a ratio (Figure 7). Cohorts that were given CONV irradiation showed a significant increase in microglial activation compared to the controls, whereas the ratio of CD68^+^/IBA1^+^ cells in FLASH-irradiated animals was intermediate and statistically similar to the controls (one-way ANOVA: F_(2,13)_ = 4.25, *p* = 0.038; *p* = 0.30: CONV vs. FLASH; *p* = 0.025: CONV vs. Control; *n* = 5–6 animals per group) (Figure 7).

### 2.6. Preservation of the Endocrine System by FLASH-RT

Cranial irradiation has been found to be responsible for marked toxicities on the hypothalamic–pituitary axis [39]. To assess the potential benefits of FLASH-RT on the pituitary function, the irradiated and control cohorts were analyzed for changes in plasma growth hormone (GH) levels as measured by ELISA at 1-week post-RT (Figure 8). Compared to non-irradiated animals, mice that received 8 Gy to the CONV-irradiated mice showed a significant (two-fold) reduction in circulating GH. Interestingly, when the same dose was delivered with FLASH-RT, no such decrease was observed, and the plasma GH levels were similar to those of the unirradiated control animals (one-way ANOVA: F_(2,21)_ = 3.13, *p* = 0.066, *p* = 0.047: CONV vs. FLASH; *p* = 0.15: CONV vs. Control; *n* = 8 animals per group). These encouraging results suggest that FLASH irradiation could prevent the development of radiation-induced toxicity in the endocrine system.

## 3. Discussion

Managing normal tissue toxicity has been a goal, a concern and a pressing clinical reality for those engaged in treating cancer with radiotherapy. The pursuit of radioprotective strategies involving improved image guided and conformal beam delivery, optimized fractionation schedules and the use of pharmacological agents have all led to tangible improvements in therapeutic outcomes. These advancements, however, have come at the expense of costly and time consuming technological and drug development programs, along with more complicated administration regimens. While the present implementation of FLASH-RT is evolving, the capability of this innovative irradiation modality to dovetail into existing hypofractionation protocols, freeze organ motion and achieve significant normal tissue sparing by a relatively straightforward change in dose rate has now provided the radiation oncology field with heretofore untold opportunities [40]. The data presented here provide further context for this promising advancement, where preclinical findings point to the potential ability to mitigate the adverse neurological consequences of radiotherapy in the treatment of MBs [3,41].

As the most common malignant childhood CNS tumor, approximately 500 new cases of MB are recorded per year in the United States [1]. Localized in the posterior fossa, they occur throughout childhood, with two identified peaks of incidence between 3–4 and 7–10 years of age [1]. Nevertheless, 15–20% of medulloblastomas are developed under 2 years of age, which complicates the therapeutic management due to the high sensitivity of the developing CNS [2]. Currently, treatment management consists of surgery and post-surgical adjuvant cranio-spinal radiation therapy and chemotherapy for children above 3 years of age [4,11,42,43,44]. Survivors suffer from a panoply of neurological complications that involve dramatic age and dose-dependent reductions in IQ [5,10,45,46] and a host of special needs in their social, emotional and educational development [47,48]. In contrast to adult malignancies of the CNS (e.g., glioblastoma multiforme, GBM), MB is more clinically tractable, albeit not without confounding complications that present challenges throughout life. The possibility of mitigating even a fraction of these neurological complications portends marked changes in subsequent clinical practice as well as lifestyle changes for survivors.

Using a juvenile mouse model, we have focused our studies on the comprehensive neurological assessment of tumor-free cohorts subjected to FLASH and CONV irradiation. While current studies need to be extended to the study of orthotopic MB tumors, we emphasize that the use of tumor free animals facilitates the reliable and reproducible quantification of behavioral performance in the absence of confounding disease, with experiments designed to specifically analyze the potential differences between the two radiation modalities. Furthermore, the lack of imaging on the experimental eRT6 FLASH irradiator precludes our ability to achieve a high degree of conformality to a specified target. Nonetheless, these analyses have revealed clearly that FLASH-RT ameliorates the wide range of radiation-induced cognitive deficits that are commonplace with CONV irradiation. The hippocampal and cortical based deficits that have been found using a comprehensive behavioral battery point to the neurocognitive sparing capabilities of FLASH-RT. While the neurological benefits of FLASH-RT remain to be defined at a precise molecular level, temporal improvements in spatial memory recognition point to the sparing of hippocampal neurogenesis.

We used a novel hippocampus-dependent memory updating task (OUL, [27]) to investigate memory updating in animals irradiated with FLASH versus CONV-RT. In the laboratory, memory is typically studied as a *de novo* experience, where a naïve animal is presented with a discrete learning event that is markedly different from its past experiences. Most real-world memories, on the other hand, are updates/additions to memories that already exist. The OUL task is a promising new tool that allows us to examine an original linear memory as well as updated information simultaneously within a single test session. We showed that while CONV-irradiated mice displayed deficits in memory updating at earlier as well as later timepoints, FLASH-irradiated animals were able to recover impaired memory updating when tested later (4 months). There are several possible explanations for this finding. One is that while the attrition and/or integration of newly born neurons in the dentate gyrus are impaired by CONV-RT, they are preserved in the FLASH-irradiated brain. This idea is supported by data demonstrating for the first time that both newly born immature (DCX_+_) and mature (NeuN^+^) neuronal populations in the hippocampal dentate gyrus were depleted only by CONV irradiation. Thus, the preservation of the neurogenic niche may promote the recovery of memory updating at the later 4-month time, and this corroborates previous findings regarding the impact of FLASH-RT on hippocampal cell division [13,16]. Furthermore, the hippocampal-cortical memory circuit, which arborizes (strengthens) as animals mature with age, may be less affected by FLASH-RT. For example, there is a direct projection from the entorhinal cortex to area CA1 of the hippocampus (temporoammonic, TA, pathway, [49,50]), effectively bypassing the first two stages of the conventional tri-synaptic circuit. The TA pathway consists of axons from the pyramidal cells of layer III of the entorhinal cortex [50,51] that terminate preferentially in the CA1 stratum lacunosum-moleculare (SLM). These axons make excitatory synapses, >90% of which are onto the spines of CA1 pyramidal cells [50,51] that play important roles in memory consolidation. Lastly, FLASH-irradiated and control mice may retain the information they learned from the initial OUL trial (at 2 months), thereby facilitating memory updating at the later time, whereas CONV-irradiated animals are simply unable to recall the first trial.

In this current study, we also utilized the NOR task, which relies on memory for familiar objects and the intrinsic preference of rodents to explore novel objects. Studies have shown that NOR depends on post-rhinal and insular cortices rather than the hippocampus [52,53,54]. Similar to our OUL findings, we see that unlike CONV-RT, FLASH-RT does not cause impairments in this task. Together, these findings suggest that FLASH irradiation has the unique ability to preserve hippocampal and perirhinal cortical circuity. In addition, FLASH irradiation did not result in behaviors that are characteristic of anxiety and depression, corroborating past behavioral results that were obtained in animals which were irradiated as adults [15]. Interestingly, FLASH animals showed similar levels of social interaction as the controls, while CONV-irradiated mice avoided interacting with the novel mouse, as shown on the SIT task. The decrements found after CONV irradiation extend to other regions, as pronounced disruptions in FE and social interaction suggest that the circuits connecting the afferent and efferent medial prefrontal circuitry with the amygdala have undergone extensive plasticity, effects that were simply not observed after FLASH-RT. Lastly, the capability of FLASH-RT to attenuate inflammation in juvenile mice corroborates much of our previous data obtained in the lungs [12] and the brains [15] of adult mice which were subjected to FLASH irradiation, and it points to the multiple mechanisms by which reductions in chronic inflammation can be beneficial in the irradiated microenvironment. In summary, the present behavioral findings obtained from irradiated juvenile mice point to the remarkable capability of FLASH-RT to ameliorate a wide range of neurocognitive complications which are associated with the CONV irradiation used in current clinical practices.

What remains to be elucidated is the precise molecular target/s for the preservation of these presumably radiosensitive targets in the CNS. While molecular studies suggest a role for neurogenesis, other studies point to caveats with this interpretation, with reasons including a minimal number of new neurons added to the dentate daily [55] and disputes regarding the occurrence of neurogenesis in higher primates altogether [56]. Regardless of the extent to which overall neurogenesis and/or reduced microgliosis contributes to the neurocognitive endpoints assessed, FLASH-RT retains the marked capability to obviate the onset of the behavioral and emotional disruptions that plague standard MB radiotherapeutic treatments.

Interestingly, hormonal changes also accompany the beneficial effects of FLASH-RT that implicate the idea of the generalized preservation of cell and/or functional viability. A dysregulation of growth hormone (GH) levels following radiation therapy has been described as dose-dependent [57,58,59] as early as 3 months post-treatment, with a permanent GH deficiency observed in more than 80% of patients one year after treatment completion [57,60]. Children with growth hormone deficiencies exhibit growth failure, with a growth deceleration following radiation therapy [57,59,61]. Moreover, treatments with GH injections do not completely compensate the endogenous lack of hormones, mainly due to the age of the patients and other factors [39]. The finding that FLASH-RT does not elicit reductions in GH levels compared to CONV-RT points to its ability to protect the pituitary gland from doses outside the target volume and the exciting possibility of forestalling growth retardation in early adults who are subjected to cranial radiotherapy. While weight differences were not observed between the irradiated cohorts, the data provide provocative insights into the additional, non-cognitive benefits that may well be obtained through the implementation of FLASH-RT for the treatment of juvenile patients.

## 4. Materials and Methods

### 4.1. Mice

Mouse experiments were approved by the Swiss (VD 3241) and American (Institutional Animal Care and Use Committee, IACUC) Ethics Committees for Animal Experimentation and performed within institutional guidelines. Pregnant C57Bl/6J female mice were purchased from Charles River Laboratories and male and female pups were weaned to 21 days of age. Mice were maintained in standard housing conditions (20 ± 1 °C; 70% ± 10% humidity; 12 h:12 h light and dark cycle) and provided ad libitum access to standard rodent chow and water.

### 4.2. Whole Brain Irradiations

Irradiation was performed using a prototype 6 MeV electron beam, LINAC, of the Oriatron type 6e (eRT6; PMB Alcen), available at Lausanne University Hospital and described previously [62]. Physical dosimetry has been extensively described and published to ensure reproducible and reliable biological studies [13,63,64,65]. This LINAC was able to produce a pulsed electron beam at a mean dose rate ranging from 0.1 Gy/s (i.e., comparable to CONV used in RT) up to 4.4 × 10^6^ Gy/s (at standard distance), corresponding to a dose, in each electron pulse, ranging from 0.01 up to 8 Gy. All FLASH irradiations were performed at an instantaneous dose rate above 4.4 × 10^6^ Gy/s (i.e., the intra-pulse dose rate). The beam parameters that were used throughout this study are included in Table 1. The irradiation settings corresponding to the prescription dose for mouse irradiations were determined by surface dose measurements on a 30 × 30 cm^2^-solid water phantom, positioned behind a graphite applicator (13.0 × 13.0 × 2.5 cm^3^) with a 1.7 cm-diameter semicircular central aperture, adapted to the size of 3 week old mice for whole brain irradiations (WBI).

All brain irradiations were performed under isoflurane anesthesia. Animals were irradiated at 3 weeks of age (post-weaning). For WBI, the mouse was positioned directly behind the graphite applicator (in contact) with the head positioned in the 1.7 cm diameter semicircular aperture in order to irradiate the whole encephalon region, while limiting the dose to the eyes, the mouth and the rest of the body. For all experiments, the mice were divided into 3 experimental groups (control, FLASH and CONV), where the FLASH and CONV groups received 8 Gy single doses (see Table 1 for irradiation parameters, *n* = 16/group).

### 4.3. Cognitive Testing

To determine the effects of CONV and FLASH dose rate irradiations on cognitive function, mice were subjected to behavioral testing at 2 and 4 months after irradiation. Early testing (2 months) was conducted over 2 weeks and included the novel objects in updated location (OUL) task, which is described in detail below. At 4 months post-RT, the same cohort of animals underwent a second round of OUL, followed by a novel object recognition (NOR) task, light-dark box test (LDB), social interaction test (SIT) and fear extinction (FE) memory test. This second timepoint of behavioral testing was done across one month. Data analysis was conducted independently and blind and is presented as the average of all trials scored for each task. For OUL and NOR, analysis was conducted on minutes 1–3 of the 5 min session since minutes 0–1 served as an acclimation phase for this task and animals showed reduced interest in object exploration at the end of the sessions (minutes 3–5).

### 4.4. Apparatus

The OUL, NOR and SIT were all conducted in a dimly lit (48 lux) test arena (30 × 30 × 30 cm) with a layer of fresh bedding that was filmed from above. For OUL testing, each box had a strip of blue-colored duct tape in the middle of one panel that served as an orienting mark. For OUL and NOR testing, which included objects, objects were cleaned with 10% ethanol at the end of each round of testing. All sessions were videotaped for the offline analysis of object exploration.

### 4.5. Objects in Updated Locations (OUL) Task

Mice were handled for two minutes per day for four days, followed by six consecutive days of habituation. Following handling and six days of habituation to the testing apparatus, mice were trained with two identical objects in specific locations (A_1_ and A_2_) for 3 days in the habituated context. The objects were plastic and were magnetically affixed 16 cm apart in the arena, and the mouse was allowed five minutes to explore the objects. After 24h, mice were given an update session in which one object was moved to a new location (A_3_). Finally, mice were given a retention test in which they were exposed to the three objects in previously experienced locations (A_1_, A_2_, A_3_) and a fourth object in a novel location A_4_. Memory for the update was inferred by comparing the exploration of the novel location A_4_ to location A_3_. Preference for the updated information was expressed as a discrimination index (DI): (tA_4_ − tA_3_)/(tA_4_ + tA_3_) × 100, where t indicates the time spent exploring the designated object. Memory for the original training information was inferred by comparing the exploration of the novel location A_4_ to locations A_1_ and A_2_. Preference for the original training information was inferred by calculating the DI to compare the exploration of the novel location A4 to objects in the original training locations A_1_: DI = (tA_4_ − tA_1_)/(tA_4_ + tA_1_) × 100.

### 4.6. Novel Object Recognition (NOR)

Similar to OUL, NOR relies on intact hippocampal, medial prefrontal cortex and perirhinal cortex function [66,67]. The NOR task evaluates episodic recognition memory by measuring the preferences of mice when investigating novel environmental changes. Briefly, mice were initially habituated to the empty arena for three days (10 min/day). The following testing day, two plastic objects (differing in color, shape and size) were magnetically affixed 16 cm apart in the arena, and the mouse was allowed five minutes to explore the objects. The mouse was returned to the home cage for five minutes while one familiar object was substituted for a novel object. The mouse was then returned to the arena for five minutes of further exploration. The DI for this task was then calculated for each mouse from these values: ((novel/total exploration time) − (familiar/total exploration time)) × 100.

### 4.7. Light-Dark Box (LDB) Test

We evaluated anxiety-like behavior with the LDB test, using the established methods [32,68,69]. The LDB arena consisted of a light compartment (30 × 20 × 27 cm, 915 lux) connected to a dark compartment (15 × 10 × 27 cm, 4 lux) via a small opening (7.5 × 7.5 cm). Thus, the LDB test contrasts the natural propensity of mice to explore new environments with their degree of anxiety to be in a well-lit space. Mice were placed in the arena for 10 min, and we recorded the time spent in each chamber and the number of transitions between compartments.

### 4.8. Social Interaction Test (SIT)

Social interaction and social avoidance behaviors were evaluated in mice using the established protocols [35,36,37]. Mice were initially each individually habituated to the well-lit (915 lux) test arena (30 × 30 cm) for two days (5 min/day). On the third day of the trial, a novel mouse of the same sex (C57BL6/J, weighing less than the test mouse) was allowed to explore freely for 10 min, prior to the test mouse being placed into the arena. The mice were allowed to explore and interact freely without barriers for 10 min, and active interaction or avoidance was recorded. Social interactions included any time the test mouse spent sniffing while in active contact with the novel animal’s snout, flank or anogenital area, mutual grooming or directed pursuit of the novel mouse. Concurrently, avoidance behavior was characterized as the time that the test mouse spent actively avoiding social interactions initiated by the novel mouse.

### 4.9. Fear Extinction Testing

To test whether mice could learn and later extinguish conditioned fear responses, we performed a series of established fear extinction (FE) assays, as adapted from [70]. Testing occurred in two similar contexts within a behavioral conditioning chamber (17.5 × 17.5 × 18 cm, Coulbourn Instruments) with steel slat floors (3.2 mm diameter slats, 8 mm spacing). For context A, the chamber was scented with a spray of 10% acetic acid in water, while in context B, the steel floor of the chamber was covered with white plastic and a spray of 10% almond extract in water was applied. Initial fear conditioning was performed in context A after mice were allowed to habituate to the chamber for two minutes. Three pairings (spaced by 120 s) of an auditory conditioned stimulus (16 kHz tone, 80 dB, lasting 120 s; CS), co-terminating with a foot-shock unconditioned stimulus (0.6 mA, 1 s; US) were presented. On the following three days of extinction training, mice were initially habituated to context B for two minutes before being presented with 20 non-US reinforced CS tones (16 kHz, 80 dB, lasting 120 s, at 5 s intervals). On the final day of fear testing, mice were presented with only three non-US reinforced CS tones (16 kHz, 80 dB, lasting 120 s) at a two-minute intertrial interval in context B. Freezing behavior was recorded with a camera mounted above the chamber and scored by an automated, video-based motion detection program (FreezeFrame, Coulbourn Instruments). FreezeFrame algorithms calculate a motion index for each frame of the video, with higher values representing greater motion. An investigator who was blinded to the experimental groups set the motion index threshold representing immobility for each animal individually, based on identifying a trough separating low values during immobility and higher values associated with motion. Motion index thresholds were set between 10 and 16 for all animals. Freezing behavior was defined as continuous bouts of 1 s or more of immobility. The percentage of time that each mouse spent freezing was then calculated for each phase of the fear response testing. For the three consecutive days of extinction training, data are presented as the averages of the first 10 non-US reinforced CS tones for each animal. Moreover, for the fear extinction test, data are presented as the averages of the three non-US reinforced CS tones for each animal.

### 4.10. BrdU Treatment and Immunohistochemistry Analyses

To assess the impact of chemotherapy on hippocampal neurogenesis, 5-bromo-2′-deoxyuridine (BrdU; 100 mg/kg intraperitoneally; Sigma-Aldrich) was administered for 6 consecutive days. One month later, animals were euthanized by intracardiac perfusion with 4% PFA (paraformaldehyde, Sigma-Aldrich). Tissues were cryo-protected (10% to 30% sucrose gradient, Sigma-Aldrich) and cryo-sectioned using a cryo-stat (Microm HN 525 NX, Thermo-Scientific, US) at a coronal thickness of 30 µm. To assess the impact of radiation exposure on the neurogenic niche, tissue were stained for immunohistochemistry (IHC), as described previously for doublecortin (DCX) and BrdU-NeuN [71,72]. Confocal analyses were carried out using multiple z-stacks that were taken at 1-μm intervals using a confocal laser-scanning microscope (Nikon Eclipse Ti-C2 interface). The individual z-sections were then analyzed using Nikon Elements software (version 4.3). The main determinant for the assessment of radiation effects on the hippocampal neurogenesis was the percentage of BrdU-positive cells co-expressing the mature neuronal marker, NeuN. The percentage of dual-labeled BrdU-NeuN, DCX and the ratio of activated (CD68^+^) to the total (IBA1^+^) microglia were each derived from 4–6 individual animals per group.

To identify mature neurons, representative sections were processed using dual immunofluorescence staining for BrdU and the neuron-specific nuclear antigen (NeuN). Serial sections taken through the caudal diencephalon were selected for staining and stored in phosphate-buffered saline (PBS, 100 mmol/L, pH 7.5, Gibco) overnight. Free-floating sections were first rinsed in tris-buffered saline (TBS, 100 mmol/L, pH 7.5, Gibco) and subjected to a BrdU pretreatment protocol using 50% formamide (made in 2× saline-sodium citrate, SSC, buffer; Sigma-Aldrich) at 68°C for 2 h and 2N HCl (at 37 °C for 45 min), followed by serum blocking (10% normal donkey serum, NDS; Jackson Immuno Research) and overnight incubation in a rat anti-BrdU solution (1:200; AbD Serotec). The sections were then treated with donkey anti-rat IgG Alexa Fluor 594 (1:200; Invitrogen) for 60 min, rinsed in PBS, then blocked in serum and incubated overnight with primary antibodies (mouse anti-NeuN, 1:200; Millipore). The following day, sections were washed with PBS and treated with a biotinylated secondary antibody (horse anti-mouse, 1:200; Vector Labs). Color development was facilitated by fluorescein (1:200 in PBS; Vector Labs). Immunostained sections were rinsed in PBS and mounted on clean Vectabond (Vector Labs)-treated slides using Slow Fade anti-fade mounting medium (Invitrogen). BrdU-positive cells were visualized using fluorescence microscopy as red and NeuN-positive cells were visualized as green.

Doublecortin (DCX) staining was performed to identify immature neurons, as previously described [72]. Sections were incubated overnight at 4 °C with a rabbit anti-DCX primary antibody (1:500; Abcam) and subsequently with biotinylated donkey anti-rabbit for one hour at room temperature. Sections were then treated in an ABC kit (Vector Laboratories) and color developed using Vector Grey (Vector Laboratories). Tissue sections were mounted on clean Superfrost Plus slides (Fisher Scientific) and counterstained using Nuclear Fast Red (Vector Laboratories). The stereologic enumeration of the immature neurons was conducted using a Nikon Eclipse microscope (TE2000, Nikon, Japan) equipped with an MBF CX9000 color digital camera, 100× (oil-immersion, 1.30NA) objective lens, 3-axis motorized stage (Ludl Electronic Products, NY) and an optical fractionated probe in the StereoInvestigator software (MBF Biosciences, v11, Vermont). The counting frame was set to 60 μm × 60 μm, and the sampling grid was set to 75 μm × 75 μm with an average mounted section thickness of 15 μm. The average Gunderson’s co-efficient of error was <0.07 ± 0.005 (mean ± SEM, *n* = 9).

For the assessment of microglia, the following primary and secondary antibodies were used: goat anti-IBA-1 (1:200, Novus, NB100-1028), rat anti-mouse CD68 (1:500, BioRad, Cat. No. 1957), donkey anti-goat or anti-mouse conjugated with Alexa Fluor 488 or 594 (Life Technologies/Invitrogen) and DAPI nuclear counterstain (Sigma-Aldrich, St. Louis, MO, USA). Two representative sections per animal, from 5–6 animals per group, were selected through the middle of the hippocampus and stained as previously described [23]. IBA-1 or CD68 positive cells were visualized under fluorescence as green against the DAPI stained nuclei (blue). Immunofluorescent sections were imaged using the Nikon Eclipse Ti C2 microscope to obtain 25 to 30 z stacks (1024 × 1024 pixels, 1 μm each), using a 20× PlanApo lens (Nikon). For the quantification of the IBA-1^+^ and CD68^+^ cells, the Imaris spot tool (v8.0, Bit Plane AG, Zürich, Switzerland) was used to detect immunostained cell bodies within the 3D deconvoluted image stacks, based on a predefined diameter and the red/green channel intensity threshold. IBA-1 and CD68 data are expressed as the ratio of the number of CD68^+^/IBA1^+^ cells relative to the unirradiated (0 Gy) controls.

### 4.11. Plasma Growth Hormone Quantification

Blood was collected from tail-vein using EDTA-coated tubes one-week post-irradiation. Cells were removed from plasma by centrifugation for 10 min at 2000× *g* at 4 °C. Supernatant was collected and stored at −80 °C. Plasma Growth Hormone (GH) titration was performed using a commercial ELISA kit (Rat/Mouse growth hormone Elisa kit EZRMGH-45K, Millipore-Sigma).

### 4.12. Statistical Analyses

Statistical analyses were carried out using GraphPad Prism (v8) software. A one-way ANOVA was used to assess the significance between the control and irradiated groups, and when the overall group effects were found to be statistically significant, a t-test with the Bonferroni correction for the multiple comparisons test was used to compare the control and FLASH groups against the CONV group. For the fear extinction experiment, conditioning day 1 (T1–T3) and extinction training days 1–3 were analyzed using two-way ANOVAs, followed by a t-test with the Bonferroni correction for multiple comparisons, with freezing time (%) as within-subjects variables and group/irradiation treatment (0 Gy controls vs. 8 Gy CONV vs. 8 Gy FLASH). This statistical test was used to make specific comparisons when significant interactions and/or main group effects were observed. Results were expressed as mean values ± SEM and all analyses considered a value of *p* ≤ 0.05 to be statistically significant.

## 5. Conclusions

As pre-clinical work strives to identify the mechanistic basis of the FLASH effect, in vivo data continue to accumulate that point to the widespread normal tissue benefits of this burgeoning new technology. These benefits are associated with an efficacious anti-tumor effect, described in several preclinical studies, a phase I-II vet clinical trial and a first patient [19,73]. Furthermore, additional work from our group has found that using a preclinical orthotopic mouse model of GBM, single and fractionated FLASH and CONV irradiation afforded equivalent tumor growth delay (Montay–Gruel, P. personal communication and [14]). While the 6 MeV electrons that were generated from the LINAC used in this study preclude the treatment of deeper tumor sites, these energies could be used under an intra-operative setting to treat certain brain or other tumors. In addition, higher energy to very high energy electron (VHEE), photon (X-rays) and proton-based irradiation modalities are currently under development for the eventual translation to the clinic. As with the advent of any new, potentially groundbreaking changes in clinical practice, there comes some well-grounded skepticism and the need for deeper data sets before such change can be completely embraced. Nonetheless, data to date from multiple groups, using different radiation modalities on various pre-clinical models, tissues organs and a single human case, have validated the “FLASH effect”. If the present findings can indeed be extended to the treatment of pediatric cancer patients afflicted with MB, as well as other CNS and non-CNS malignancies, then lifelong complications stemming from their treatments may one day be lessened, if not eliminated. This provides ample justification to aggressively pursue further studies in order to deepen our understanding of radiation’s dose rate effects.

## Figures and Tables

**Figure 1 cancers-12-01671-f001:**
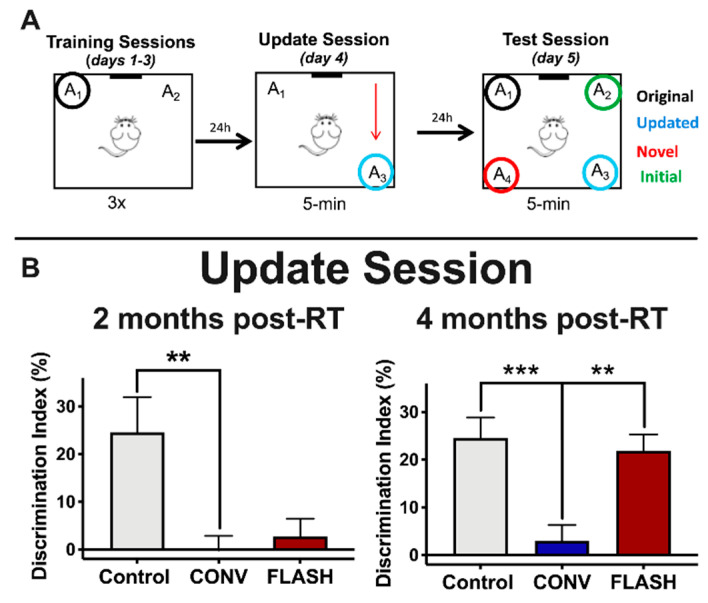
Animals exposed to ultra-high dose rate (FLASH) radiotherapy (RT) successfully perform memory updating in the objects in updated locations (OUL) task with time, but animals exposed to conventional dose rate (CONV) irradiation do not. (**A**) Experimental design. (**B**) Update session behavior. At 2 months post-RT, both CONV (8 Gy) and FLASH-irradiated (8 Gy) mice show no memory update (significantly lower discrimination index (DI) values compared to controls). At 4 months post-RT, however, FLASH-irradiated (8 Gy) animals successfully learn during the update session but CONV-irradiated animals do not, indicating that FLASH-irradiated animals are able to learn similarly to controls with time (CONV-irradiated animals show no update memory at both timepoints). Mean ± SEM (*n* = 15–16 per group); *p*-values were compared against CONV and derived from one-way ANOVA followed by t-test with Bonferroni’s correction for multiple comparisons. ** *p* < 0.01, *** *p* < 0.001.

**Figure 2 cancers-12-01671-f002:**
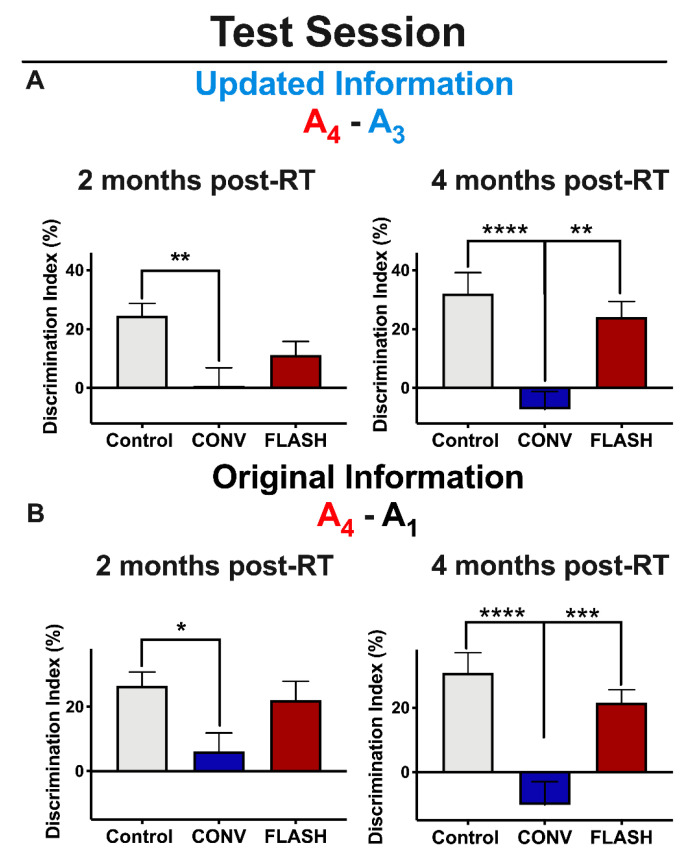
Mice irradiated with FLASH-RT acquire both the updated memory and the original memory during the objects in updated locations (OUL) test session with time, whereas mice irradiated with CONV-RT remain impaired. (**A**) Updated information during the test session. Left: at 2 months post-RT, both CONV (8 Gy) and FLASH-irradiated (8 Gy) animals fail to acquire the updated information. (**A**) Right: at 4 months post-RT, however, FLASH-irradiated (8 Gy) animals successfully learn the updated information on test day, but CONV-irradiated animals do not. (**B**) Original information during the test session. Left: at 2 months post-RT, both CONV (8 Gy) and FLASH-irradiated animals (8 Gy) fail to acquire the original information. Right: at 4 months post-RT, FLASH-irradiated (8 Gy) animals successfully learn the original information on test day, but CONV-irradiated animals do not. Mean ± SEM (*n* = 15–16 per group); *p*-values were compared against CONV and derived from one-way ANOVA followed by t-test with Bonferroni’scorrection for multiple comparison: * *p* < 0.05, ** *p* < 0.01, *** *p* < 0.001, **** *p* < 0.0001.

**Figure 3 cancers-12-01671-f003:**
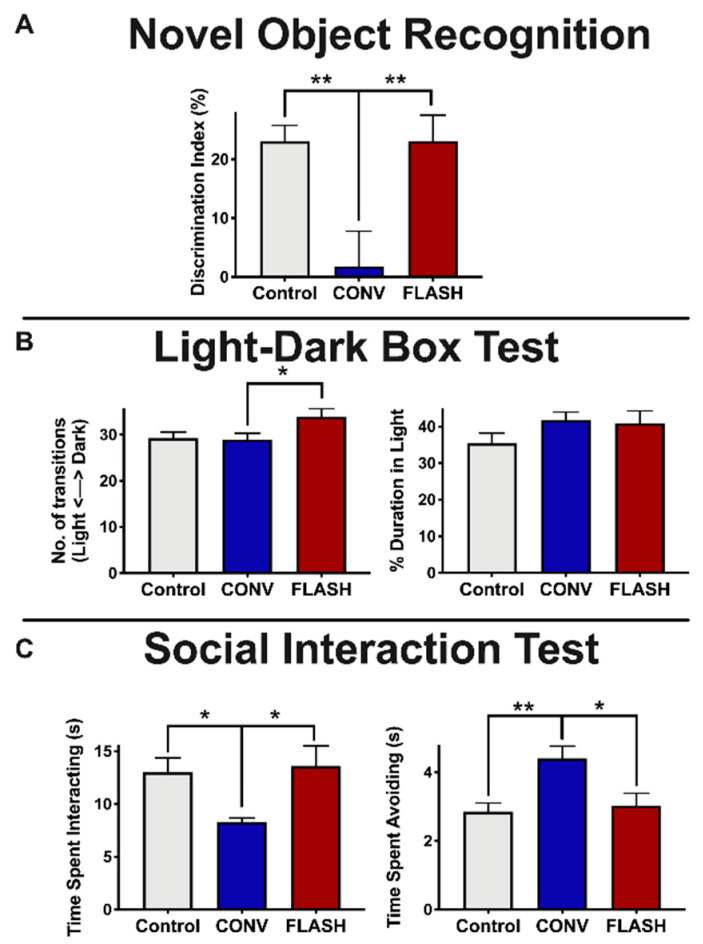
FLASH-RT minimizes radiation-induced novel object memory impairments as well as anxiety-like behaviors. Mice were tested for cognitive function using novel object recognition (NOR; **A**), light-dark box test (LDB; **B**) and social interaction test (SIT; **C**) at 4 months post-RT. (**A**) NOR: Animals exposed to FLASH irradiation (8 Gy) have statistically indistinguishable high discrimination index (DI) scores relative to controls, indicating a preference for the novel object. (**B**) Left: mice exposed to FLASH irradiation showed significantly more transitions between the light and dark regions of the LDB compared with CONV-irradiated mice. Right: no group differences were observed in percentage of time spent in the light compartment. (**C**) Left: SIT testing reveals a significant reduction in time spent interacting in the CONV-irradiated animals compared to control and FLASH groups. Right: Furthermore, SIT testing showed a significant increase in avoidance behavior compared with controls and FLASH-irradiated mice. Mean ± SEM (*n* = 14–16 per group); *p*-values were compared against CONV and derived from one-way ANOVA followed by t-test with Bonferroni’s correction for multiple comparison. * *p* < 0.05, ***p* < 0.01.

**Figure 4 cancers-12-01671-f004:**
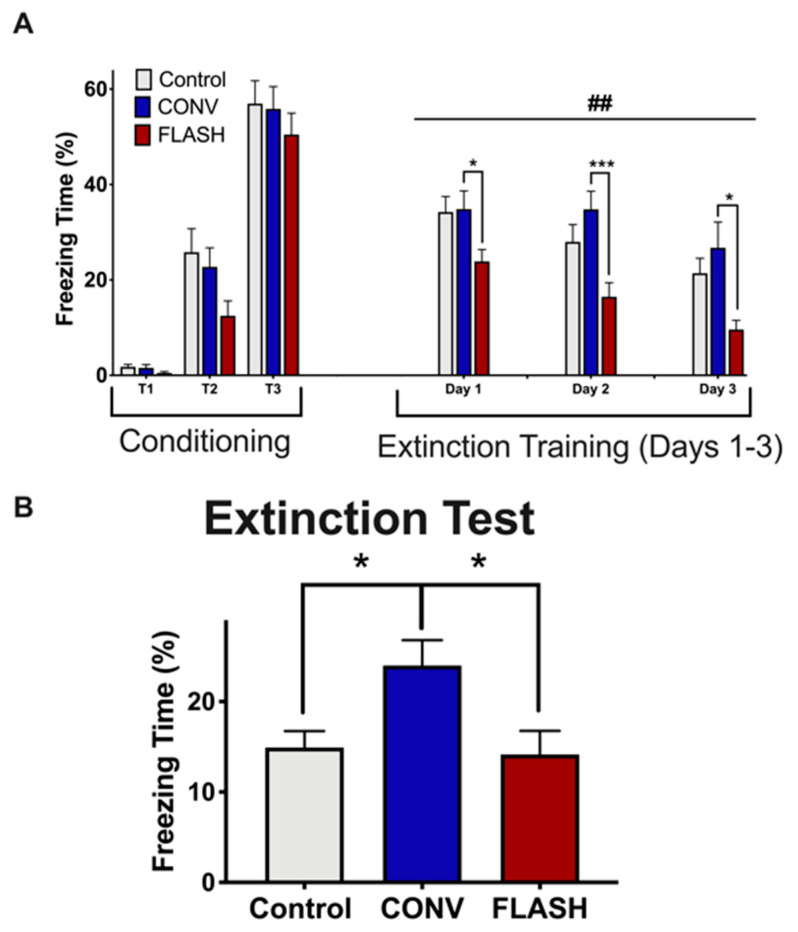
FLASH-RT does not induce deficits in fear extinction memory. (**A**) Exposure to either irradiation modality did not impair the acquisition of conditioned fear (three tone-shock pairings). All mice showed a gradual decrease in freezing behavior over extinction sessions (tone only); however, the time spent freezing was significantly greater for the mice irradiated with CONV-RT (8 Gy) compared with the controls or FLASH-irradiated (8 Gy) animals. (**B**) Control and FLASH-irradiated animals successfully abolished fear memory compared to the CONV group. Mean ± SEM (*n* = 15–16 per group); ## indicates significant main group effect, *p* < 0.01. *p* values for (**A**) were derived from two-way ANOVA followed by *t*-test with Bonferroni correction for multiple comparison. *p* values for (**B**) were compared against CONV and derived from one-way ANOVA followed by the Bonferroni’s multiple comparision test: * *p* < 0.05, *** *p* < 0.001.

**Figure 5 cancers-12-01671-f005:**
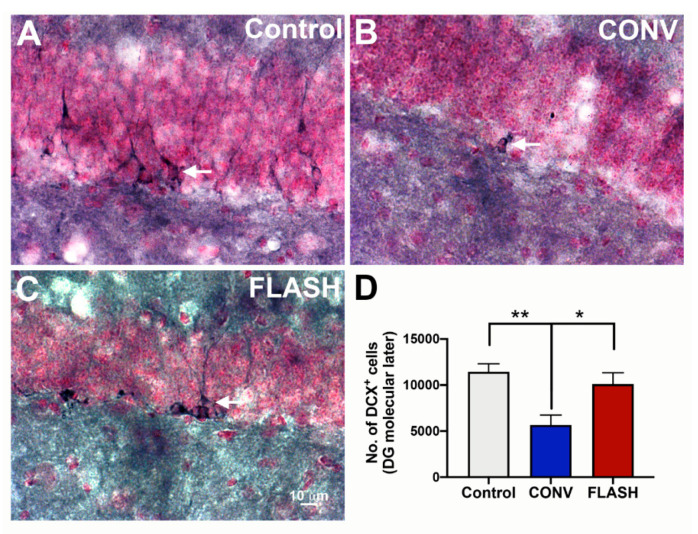
FLASH-RT preserves the neurogenic niche. Immature neurons (doublecortin, DCX^+^) were quantified using unbiased stereology 4 months post-irradiation of juvenile mice. Compared to controls (**A**) and FLASH-RT (**B**), CONV-RT (**C**) leads to a significant decline in the DCX^+^ immature neurons (**D**). Mean ± s.e.m. (*n* = 4 animals per group), *p* values were compared against CONV and derived from ANOVA and Bonferroni’s multiple comparisons test. ** *p* < 0.01, * *p* < 0.05. Scale bar: 10 μm, (**A**–**C**).

**Figure 6 cancers-12-01671-f006:**
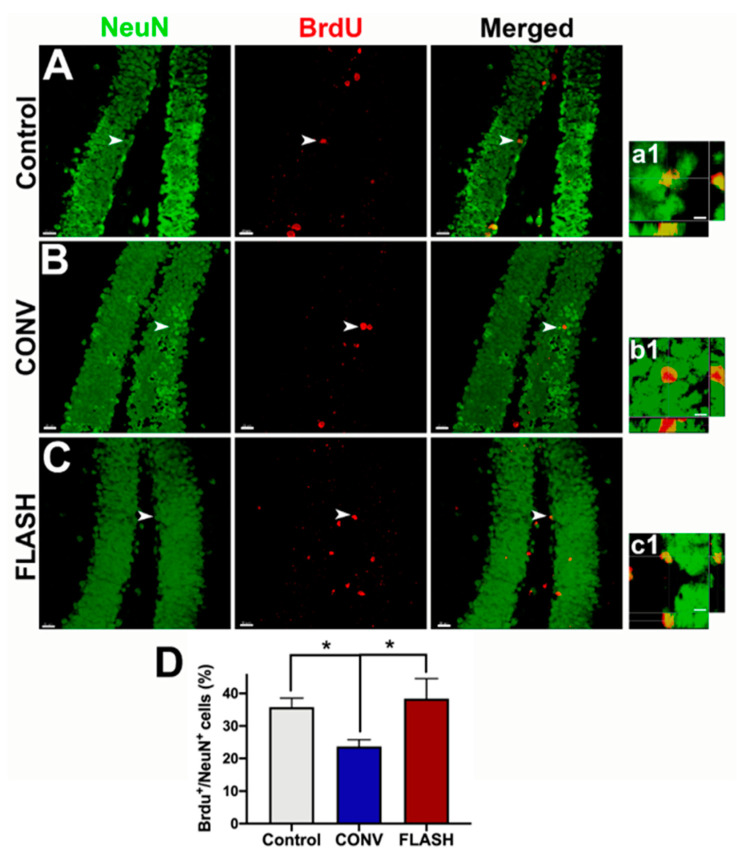
FLASH-RT preserves neurogenesis. Hippocampal neurogenesis was quantified using BrdU-NeuN dual-immunofluorescence in the dentate gyrus four months following irradiation. Mice received BrdU injections one month prior to tissue collection. Compared to the controls (**A**,**a1**), mice irradiated with conventional dose rate RT (CONV; **B**,**b1**) showed a significant decline in neurogenesis, as indicated by the reduced numbers of percentage of BrdU^+^ cells (red) differentiating into mature neurons (green, NeuN; **C**,**c1**,**D**). FLASH-irradiated mice retained significantly greater numbers of dual-labeled cells (BrdU^+^-NeuN^+^), similar to the control levels (**D**). Orthogonal z stacks (**a1**–**c1**) for the representative BrdU^+^-NeuN^+^ cells (arrows) are shown for each group (**A**–**C**). Mean ± SEM (*n* = 4–6 animals per group), *p* values were compared against CONV and derived from ANOVA and Bonferroni’s multiple comparisons test. * *p* < 0.05. Scale bars: 20 µm (**A**–**C**); 5 µm, (**a1**–**c1**).

**Figure 7 cancers-12-01671-f007:**
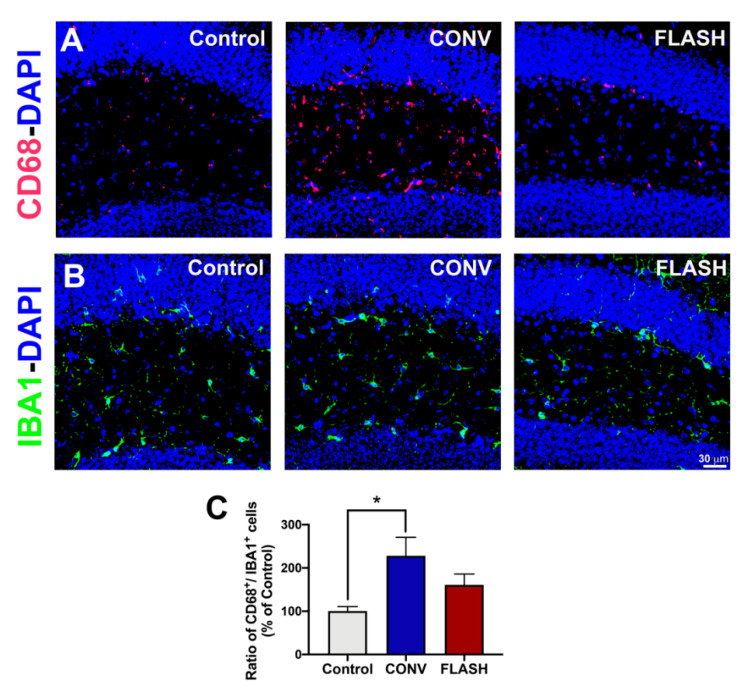
FLASH-RT does not induce microglial activation. Quantification of microglial immunoreactivity using the activated (CD68^+^; **A**) and pan-microglial (IBA1^+^; **B**) markers shows a higher ratio of CD68^+^/IBA1^+^ cells (**C**), indicating microglial activation in the CONV group compared to controls, with intermediate levels found in FLASH-irradiated mice. Mean ± SEM (*n* = 5–6 animals per group), *p* values were compared against CONV and derived from ANOVA and Bonferroni’s multiple comparisons test. * *p* < 0.05. Scale bar: 30 μm, (**A**,**B**).

**Figure 8 cancers-12-01671-f008:**
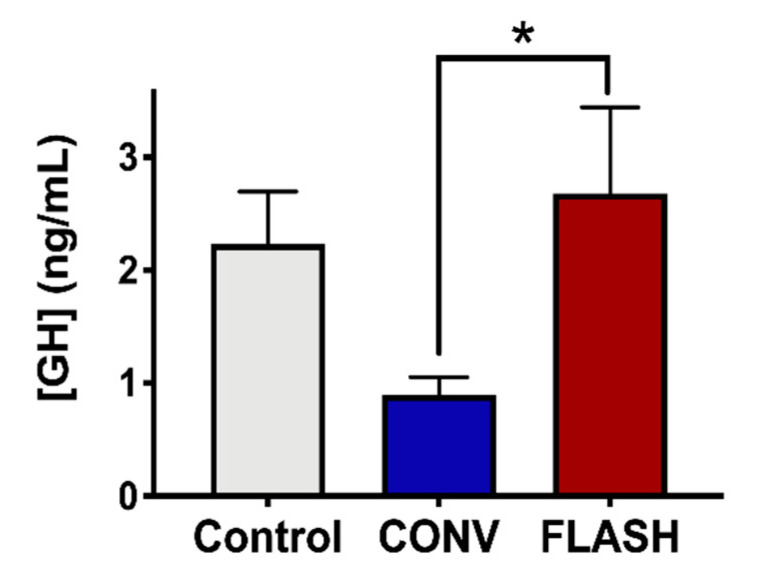
FLASH-RT preserves growth hormone (GH) levels. Plasma samples were collected 1 week after treatment from control non-irradiated animals and animals irradiated with 8 Gy at the age of 3 weeks. GH concentration in plasma was quantified by the ELISA method. Data clearly show that, compared to CONV-irradiated mice, FLASH-irradiated mice expose a higher level of plasmatic GH, comparable to control levels. Mean ± SEM (*n* = 8 animals/group), *p*-values derived from ANOVA followed by t-test with Bonferroni correction for multiple comparison. * *p* < 0.05.

**Table 1 cancers-12-01671-t001:** Irradiation parameters.

Delivery Mode	Prescribed Dose (Gy)	Graphite Applicator Type and Size (mm)	Beam Parameters						
			Source-to-Surface Distance (mm)	Pulse Repetition Frequency (Hz)	Pulse Width (µs)	Number of Pulses	Treatment Time (s)	Mean Dose Rate (Gy/s)	InstantanEous Dose Rate (Gy/s)
**CONV**	8	Semicircular ∅17	798	10	1.0	1033	103.2	0.077	7.7 × 10^3^
FLASH	8	Semicircular ∅17	383	100	1.8	1	1.8 × 10^−6^	4.4 × 10^6^	4.4 × 10^6^

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
