# Peer review of "Neuroprotection of Radiosensitive Juvenile Mice by Ultra-High Dose Rate FLASH Irradiation"

_cancers, 2020, doi:10.3390/cancers12061671_

Round 1

Reviewer 1 Report

This manuscript presents data showing how FLASH radiation is cognitive sparing to the juvenile mouse brain compared to the equivalent dose of conventional radiation. This is similar to a previous manuscript from the same group. However, in this manuscript juvenile mice (3 weeks old) were irradiated and in the previous study cognitive sparing of FLASH-RT was examined in older mice (8 weeks old). It is important to establish that the FLASH-RT is cognitive sparing in the juvenile brain when considering it for treatment of brain cancer in tumors in children.

There are a few points I would like to have addressed:

1. Although initially it is required to show effect on the brain without a tumor, it is important to demonstrate that a cognitive sparing dose of FLASH-RT is also able to effectively treat the tumor. If it is also “sparing” the tumor to the same degree it will be of no benefit. Although I appreciate that this will be addressed in future studies, I feel that some comment addressing this issue should be in the manuscript.

2. Considering the dose rate is important for the FLASH-RT, the dose rate for both FLASH and CONV should be clearly mentioned in the results or discussion as well as in the Material and Methods.

3. In the material and methods it was stated that electrons were used, but it was not mentioned or discussed elsewhere. It should be mentioning that electrons were used and how this could be (or could not be) used to treat brain cancer patients. 6 MV photons are able to penetrate the skull and deliver dose to the tumor, but electrons may not be able to penetrate far enough. Please comment on how you would plan to treat a human tumor with FLASH-RT using electrons.

Author Response

Response to Reviewers in « Blue »

Reviewer 1

This manuscript presents data showing how FLASH radiation is cognitive sparing to the juvenile mouse brain compared to the equivalent dose of conventional radiation. This is similar to a previous manuscript from the same group. However, in this manuscript juvenile mice (3 weeks old) were irradiated and in the previous study cognitive sparing of FLASH-RT was examined in older mice (8 weeks old). It is important to establish that the FLASH-RT is cognitive sparing in the juvenile brain when considering it for treatment of brain cancer in tumors in children.

There are a few points I would like to have addressed:

  1. Although initially it is required to show effect on the brain without a tumor, it is important to demonstrate that a cognitive sparing dose of FLASH-RT is also able to effectively treat the tumor. If it is also “sparing” the tumor to the same degree it will be of no benefit. Although I appreciate that this will be addressed in future studies, I feel that some comment addressing this issue should be in the manuscript.

Thank you for this comment, in fact we previously published results showing isoefficacy of FLASH and CONV irradiation on brain tumors (see ref. 74) and have a present manuscript in revision (Clinical Cancer Research) demonstrating isoefficiency between FLASH and CONV following single or hypofractionated irradiation (albeit with an orthotopic GBM tumor model).  This has been added to the conclusion section at your request (see page 24).

  1. Considering the dose rate is important for the FLASH-RT, the dose rate for both FLASH and CONV should be clearly mentioned in the results or discussion as well as in the Material and Methods.

As highlighted by the reviewer, reporting the  dose rate is  important but is more complex than reporting a single value. As mentioned in the introduction, several physics parameters need to be considered and reported.  In regard to this observation, we included Table 1 that summarizes all the physics parameters used for FLASH and CONV irradiations in the present manuscript. The instantaneous dose rate (or intra-pulse dose rate) needs to be  distinguished from the mean dose rate, this is explained in the Introduction, to provide the upmost clarity and accuracy regarding the physics parameters required to produce the FLASH effect.  Other parameters such as frequency, pulse width, number of pulsesmay be as important as the dose rate itself. Defining the physics parameters required to produce the FLASH effect is an entire new field of investigation and is outside the scope of the present article. In the present work, we investigated the response of the rodent juvenile brain to ultra-high dose rate irradiation using physics parameters already validated and known to produce the FLASH effect.

  1. In the material and methods it was stated that electrons were used, but it was not mentioned or discussed elsewhere. It should be mentioning that electrons were used and how this could be (or could not be) used to treat brain cancer patients. 6 MV photons are able to penetrate the skull and deliver dose to the tumor, but electrons may not be able to penetrate far enough. Please comment on how you would plan to treat a human tumor with FLASH-RT using electrons.

We agree with the reviewer that the 6 MeV LINAC used in this study could not be used to treat human brain-tumors, due to the low in-depth penetration of low-energy electrons. This is why we added in the conclusion section, a statement highlighting the fact that other irradiation modalities using X-rays, protons, VHEE or IORT are under development to circumvent this limitation.

Reviewer 2 Report

The study by Alaghband et al. is highly interesting. Radiobiological data on normal tissue effects following FLASH irradiation compared to conventional irradiation is requested to elucidate the likely sparing effect when compared with conventional irradiation. The present in vivo study on sparing of neurological effects following FLASH-RT and CONV exposure of juvenile murine brain is herewith a significant contribution to increase knowledge about FLASH-RT. The data is very interesting for radiobiologists and important for backing for clinical application of FLASH-RT in pediatric patients. The experimental approach is sound, the results are convincing, the presentation of the data is clear. Despite, this reviewer has some remarks and critical points, which are listed in detail below.

Specific comments

Title: the study is on sparing of neuronal tissue, not radioprotection. This reviewer would therefor suggest “Neurosparing” instead of “Neurprotection”

Abstract: Please be clear that this is an experimental study using a juvenile mice brain model! The word “mice” is not mentioned but should be mentioned in the abstract.

Introduction / Results:

My main criticism regards the experimental in vivo model and the follow-up time.

In the introduction the authors are listing a number of deficits like memory, attention, mood disorders etc., with associated pathologies such as vascular abnormalities, decrease in vascular density, demyelination ( please notice: you mention “myelination”), neuroinflammation, all long- term pathological effects.

My point is therefore: The validity of the model. How representative is the juvenile brain model you studied for the pediatric clinical situation with late, irreversible, radiation-induced side effects – appearing many years after therapy? In the discussion you write: “Using a well-defined juvenile mouse model”, but how well is this model defined, how representative is the model for the clinical situation?  

Please comment on the 4 months follow-up period. To my knowledge, this period is too short for evaluation of late adverse neurological and pathological effects of radiation – with long latent period of many months to years-, even in juvenile mice. Your explanation of better neurogenesis recovery following FLASH-RT – by better sparing of the neurogenic niche at the 4 months time point - vs. CONV is not completely convincing. Wouldn’t it be possible that the CONV irradiated animals – after a relative small single dose of 8 Gy - will show delayed recovery, which could only be assessed after prolonged follow-up time?

It would be very interesting to look at other pathological effects than preservation of the neurogenic niche in the murine brain. Did you evaluate / find any indications for vascular abnormalities, neuroinflammation, early signs of demyelination? Please comment. And if so, or not, please add those data to the results section of your ms., since “well-grounded skepticism and the need for more deeper data sets“ is needed before FLASH-RT will be completely embraced in the clinic (= your conclusion text).

Regarding the growth hormone experiments, this reviewer wondered the choice of the one week post-treatment evaluation point. In the discussion you mention that GH dysregulation following irradiation has been reported in a dose-dependent way “as early as 3 months post-treatment”. What is the rationale for selecting one week – is this representative for the clinical situation at 3 months? - , and did you repeat measurements at a later time point – with could indicate a delayed effect of CONV vs. FLASH-RT?  Please comment.

Clear figures.

Materials and Methods are well described.

Discussion is to the point, but mightbe shortened.

Literature is up to date.

Author Response

Reviewer 2

The study by Alaghband et al. is highly interesting. Radiobiological data on normal tissue effects following FLASH irradiation compared to conventional irradiation is requested to elucidate the likely sparing effect when compared with conventional irradiation. The present in vivo study on sparing of neurological effects following FLASH-RT and CONV exposure of juvenile murine brain is herewith a significant contribution to increase knowledge about FLASH-RT. The data is very interesting for radiobiologists and important for backing for clinical application of FLASH-RT in pediatric patients. The experimental approach is sound, the results are convincing, the presentation of the data is clear. Despite, this reviewer has some remarks and critical points, which are listed in detail below.

Specific comments

Title: the study is on sparing of neuronal tissue, not radioprotection. This reviewer would therefor suggest “Neurosparing” instead of “Neurprotection”

We prefer to keep our title as is.

Abstract: Please be clear that this is an experimental study using a juvenile mice brain model! The word “mice” is not mentioned but should be mentioned in the abstract.

Thank you – this has been corrected

Introduction / Results:

My main criticism regards the experimental in vivo model and the follow-up time.

In the introduction the authors are listing a number of deficits like memory, attention, mood disorders etc., with associated pathologies such as vascular abnormalities, decrease in vascular density, demyelination ( please notice: you mention “myelination”), neuroinflammation, all long- term pathological effects.

My point is therefore: The validity of the model. How representative is the juvenile brain model you studied for the pediatric clinical situation with late, irreversible, radiation-induced side effects – appearing many years after therapy? In the discussion you write: “Using a well-defined juvenile mouse model”, but how well is this model defined, how representative is the model for the clinical situation? 

Using young rodents (~2-3 weeks of age) is an accepted means for approximating the age of pediatric human patients, where MB typically present from 3-13 years of age (mean age - 8 yr).  Depending on what reference you select (doi.org/10.1016/j.lfs.2015.10.025; doi: 10.1289/ehp.00108s3511; doi: 10.1016/j.pneurobio.2013.04.001) one can approximate that a 3-week old rodent is equivalent to the toddler brain (3-4 years of age).  In the end, there is really no consensus on the “best” age for a “pediatric model”, which must depend on the endpoints selected.  Our selection of 3 weeks, was targeted based on practicality of performing irradiations, and our desire to irradiate juvenile mice at a very young and highly radiosensitive “human age” equivalent – an age in which children do routinely manifest MB.   We can however, modify out statement to eliminate our usage of “Well-defined” as this can be construed to be an overreach (see page 14).

Please comment on the 4 months follow-up period. To my knowledge, this period is too short for evaluation of late adverse neurological and pathological effects of radiation – with long latent period of many months to years-, even in juvenile mice. Your explanation of better neurogenesis recovery following FLASH-RT – by better sparing of the neurogenic niche at the 4 months time point - vs. CONV is not completely convincing. Wouldn’t it be possible that the CONV irradiated animals – after a relative small single dose of 8 Gy - will show delayed recovery, which could only be assessed after prolonged follow-up time?

We have published extensively on the normal tissue complications associated with cranial irradiation in adult mouse and rat models.  Our past work has shown that following conventional irradiation, rodents remain neurologically impaired over the course of 1-9 months, and that recovery from that damage, at a neurocognitive, structural and inflammatory standpoint do not occur in the absence of a specific intervention.

  1. Acharya, M.M., Martirosian, V, Christie, L-A. and Limoli, C.L. Long-term cognitive effects of human stem cell transplantation in the irradiated brain. J. Radiat. Biol., 90(9):816-20 (2014).
  2. Acharya, M.M., Rosi, S., Jopson, T and Limoli, C.L. Human neural stem cell transplantation provides long-term restoration of neuronal plasticity in the irradiated hippocampus. Cell Transplantation, 24:691-702 (2015).
  3. Acharya, M.M., Martirosian, V., Christie, L-A., Riparip. L., Strnadel, J., Parihar, V.K. and Limoli, C.L. Defining the optimum window for cranial transplantation of human iPS-derived cells to ameliorate radiation-induced cognitive impairment. Stem Cells Transl. Med. 4(1):74-83 (2015).

While it is possible that animals given a dose of 8 Gy – might recover, I’m not aware of any evidence in the literature that supports that possibility.  In the end, one can always undertake studies using longer follow up times, but based on past and present data, recovery from irradiation if it does transpire, has not been found under current experimental paradigms.

Lastly, I believe the inclusion of our new data directly addresses your concern regarding neurogenesis after FLASH-RT (refer to Figure 6). 

It would be very interesting to look at other pathological effects than preservation of the neurogenic niche in the murine brain. Did you evaluate / find any indications for vascular abnormalities, neuroinflammation, early signs of demyelination? Please comment. And if so, or not, please add those data to the results section of your ms., since “well-grounded skepticism and the need for more deeper data sets“ is needed before FLASH-RT will be completely embraced in the clinic (= your conclusion text).

These are all valid points which can be addressed in 2 ways:

First, we have added two additional figures, one showing that FLASH-RT spared neurogenesis in the hippocampal dentate gyrus (Figure 6), and that FLASH-RT minimizes microgliosis in the brain compared to CONV (Figure 7). 

Second, your request for further additional endpoints is welcome, but we point out that we can only undertake so much with any given cohort.  The following manuscripts (currently under review) will likely address the majority of your inquiries regarding additional neuroinflammatory endpoints and vascular abnormalities:

Montay-Gruel, P., Markarian, M., Allen, B.D., Baddour, J.D., Giedzinski, E., Jorge, P.G., Petit, B., Bailat, C., Vozenin, M-C., Limoli, C.L. and Acharya, M.M. Ultra-high dose rate FLASH irradiation limits reactive gliosis in the brain Submitted Radiat. Res. (2020).

Allen, B.D., Acharya, M.M., Montay-Gruel, P., Jorge, P.G., Bailat, C., Petit, B., Vozenin, M-C. and Limoli, C.L. Ultra-high dose rate FLASH irradiation spares CNS vascular damage in mice. Submitted Radiat. Res. (2020).

Investigations regarding myelination are currently underway in collaboration with Dr. Dara Dickstein, who is an expert in electron microscopy; these data sets are too preliminary to report. 

Regarding the growth hormone experiments, this reviewer wondered the choice of the one week post-treatment evaluation point. In the discussion you mention that GH dysregulation following irradiation has been reported in a dose-dependent way “as early as 3 months post-treatment”. What is the rationale for selecting one week – is this representative for the clinical situation at 3 months? - , and did you repeat measurements at a later time point – with could indicate a delayed effect of CONV vs. FLASH-RT?  Please comment.

GH dysregulation has been observed in human patients as early as 3 months post treatment. In our study, mice were irradiated at 3 weeks of age and sampled 1-week thereafter as you indicated.  Once animals reach adulthood, (10-12 weeks of age) drops in GH levels confound measurements.  Prior experience in our laboratory showed that GH levels at 16 weeks of age were too low to observe differences between control and irradiated cohorts.  This reality drove the selection of our early time point (1-week post-RT), since relatively higher GH levels in controls form the basis of more reliably GH quantifications, necessary to properly validate potential differences between the irradiation modalities.

Clear figures.

Materials and Methods are well described.

Discussion is to the point, but mightbe shortened.

Literature is up to date.

Reviewer 3 Report

In this study the authors test the effect of FLASH RT compared to conventional delivered radiation on juvenile mice in order to attempt to model radiation delivered to the developing brain such as with medulloblastoma. Juvenile mice without tumors were treated with either protocol then assessed at 2 and 4 months with a battery of tests including objects in updated locations, novel object recognition, light-dark box test and social interaction test. The FLASH treated mice demonstrated less significant impairment by these measures. Further evidence suggests that hippocampal neural progenitors detected by double cortin staining were more prevalent in the FLASH RT mice. Finally growth hormone levels were significantly higher in FLASH RT mice. This preclinical data suggests less functional side effects, less damage to neural progenitors and less damage to the hypothalamic-pituitary axis. The manuscript is well written and the data clearly presented. I do question whether this journal is the correct audience for this manuscript. 

Author Response

Reviewer 3

In this study the authors test the effect of FLASH RT compared to conventional delivered radiation on juvenile mice in order to attempt to model radiation delivered to the developing brain such as with medulloblastoma. Juvenile mice without tumors were treated with either protocol then assessed at 2 and 4 months with a battery of tests including objects in updated locations, novel object recognition, light-dark box test and social interaction test. The FLASH treated mice demonstrated less significant impairment by these measures. Further evidence suggests that hippocampal neural progenitors detected by double cortin staining were more prevalent in the FLASH RT mice. Finally growth hormone levels were significantly higher in FLASH RT mice. This preclinical data suggests less functional side effects, less damage to neural progenitors and less damage to the hypothalamic-pituitary axis. The manuscript is well written and the data clearly presented. I do question whether this journal is the correct audience for this manuscript.

Thank you for these comments, and we do point out that this journal is sponsoring a special issue on “Advances in Experimental Radiotherapy”.

Round 2

Reviewer 1 Report

In my opinion, this manuscript is acceptable for publishing in its present form.

Reviewer 2 Report

Thanks to the authors for the detailed and convincing answers to my comments and suggestions, and accordingly made adaptions and revisions in the original ms. No further comments. 

Reviewer 3 Report

It has sufficient scientific merit.